# Priming of Colorectal Tumor-Associated Fibroblasts with Zoledronic Acid Conjugated to the Anti-Epidermal Growth Factor Receptor Antibody Cetuximab Elicits Anti-Tumor Vδ2 T Lymphocytes

**DOI:** 10.3390/cancers15030610

**Published:** 2023-01-18

**Authors:** Jordi Leonardo Castrillo Fernandez, Roberto Benelli, Delfina Costa, Alessio Campioli, Sara Tavella, Maria Raffaella Zocchi, Alessandro Poggi

**Affiliations:** 1Molecular Oncology and Angiogenesis Unit, IRCCS Ospedale Policlinico San Martino, 16132 Genoa, Italy; 2Cellular Oncology Unit, IRCCS Ospedale Policlinico San Martino, 16132 Genova, Italy; 3Department of Experimental Medicine, University of Genoa, 16132 Genoa, Italy; 4Division of Immunology, Transplants and Infectious Diseases, IRCCS San Raffaele Scientific Institute, 20132 Milan, Italy

**Keywords:** tumor associated fibroblasts, antibody-drug conjugate, colorectal cancer, epidermal growth factor receptor, gammadelta T lymphocytes, amino-bis-phosphonates

## Abstract

**Simple Summary:**

Different cells present in the tumor microenvironment can deeply influence both cancer spreading and the success of therapy in solid tumors, including colorectal carcinoma (CRC). In particular, tumor-associated fibroblasts (TAF) exert potent immunosuppressive effects leading to the impairment of anti-tumor surveillance and interfering with the function of therapeutic antibodies. Herein, we show that CRC-derived TAF can be turned by zoledronic acid (ZA), in soluble form or as antibody-drug conjugate (ADC), into efficient stimulators of anti-tumor lymphocytes. The ADC, made of the anti-EGFR cetuximab (Cet), used in CRC therapy, and ZA (Cet-ZA) can induce the proliferation of lymphocytes that become able to kill both tumor cells and TAF. The double result is a direct anti-cancer effect and the neutralization of the inhibitory activity exerted by its stroma. The major advantage of the Cet-ZA ADC formulation is the precise delivery of ZA to EGFR^+^ cells, targeting TAF (potentially immunosuppressive), besides CRC cells.

**Abstract:**

Tumor-associated fibroblasts (TAF) exert immunosuppressive effects in colorectal carcinoma (CRC), impairing the recognition of tumor cells by effector lymphocytes, including Vδ2 T cells. Herein, we show that CRC-derived TAF can be turned by zoledronic acid (ZA), in soluble form or as antibody-drug conjugate (ADC), into efficient stimulators of Vδ2 T cells. CRC-TAF, obtained from patients, express the epidermal growth factor receptor (EGFR) and the butyrophilin family members BTN3A1/BTN2A1. These butyrophilins mediate the presentation of the phosphoantigens, accumulated in the cells due to ZA effect, to Vδ2 T cells. CRC-TAF exposed to soluble ZA acquired the ability to trigger the proliferation of Vδ2 T cells, in part represented by effector memory cells lacking CD45RA and CD27. In turn, expanded Vδ2 T cells exerted relevant cytotoxic activity towards CRC cells and CRC-TAF when primed with soluble ZA. Of note, also the ADC made of the anti-EGFR cetuximab (Cet) and ZA (Cet-ZA), that we recently described, induced the proliferation of anti-tumor Vδ2 T lymphocytes and their activation against CRC-TAF. These findings indicate that ZA can educate TAF to stimulate effector memory Vδ2 T cells; the Cet-ZA ADC formulation can lead to the precise delivery of ZA to EGFR^+^ cells, with a double targeting of TAF and tumor cells.

## 1. Introduction

It is widely accepted that the tumor microenvironment (TME) can deeply influence the onset and development of cancers. In particular, mesenchymal stromal cells (MSC) are known to exert potent immunosuppressive effects leading to the downregulation of immune cell function and impairing anti-tumor immune responses [1,2,3]. Indeed, the interaction between MSC isolated from healthy bone marrow and effector lymphocytes can lead to the suppression of T cell proliferation and cytotoxic activity [4,5,6]. MSC present within the tumor, are mostly tumor-associated fibroblasts (TAF), myofibroblasts, pericytes and adipocytes [6,7]. MSC also populate healthy tissues; in a normal gut they are a structural component of intestinal crypts and regulate epithelial cell proliferation [8,9]. According to these characteristics, MSC are considered main actors of carcinogenesis [10,11]. In colorectal cancer (CRC), TAF are known to actively contribute to imbalance anti-cancer surveillance towards immunosuppression and consequent tumor growth [12,13,14]. Thus, both MSC and TAF are considered attractive targets for anti-tumor therapies aimed to modify TME and promote immunocompetent cell function [15,16,17]. As an example, fibroblast activation protein has been proposed as a component of an anti-tumor vaccine resulted as efficient in a murine model, leading to tumor infiltration of CD8^+^ T cells, reduction of TAF number and inhibition of the recruitment of immunosuppressive cells within the tumor [16].

We have reported that CRC-TAF can impair the recognition of CRC by anti-tumor effector cells [18,19,20]; among them are Vδ2 T lymphocytes, a circulating T cell population recruited to the tumor site in several cancers, including CRC [21,22,23]. These lymphocytes can be triggered by zoledronic acid (ZA), that is taken up by antigen-presenting cells such as monocytes or cancer cells, and induce accumulation of pyrophosphate antigens (PA) through interference with the mevalonate pathway [24,25]. Subsequently, PA are exposed on the CRC cell surface by the butyrophilin subfamily 3 member A1 and subfamily 2 member A1 (BTN3A1/BTN2A1) and presented to the T cell receptor (TCR) of Vδ2 T cells [26,27]. For this reason, aminobisphosphonates (N-BPs), including ZA, have been proposed as useful tools in cancer immunotherapy [28,29,30]. We also demonstrated that N-BPs, acting on the mevalonate biosynthetic pathway, can regulate MSC-induced T-cell suppression and B-lymphocyte survival [31].

More importantly, we reported that the exposure of CRC to ZA, soluble or nanoformulated, leads to the expansion of Vδ2 T lymphocytes with effector memory phenotype and sensitizes cancer cells to γδ T cell-mediated cytotoxicity [19,32,33]. Furthermore, BTN3A1 is detected in CRC at the tumor site, on epithelial and stromal cells, often close to areas infiltrated by Vδ2 T lymphocytes, making conceivable the use of N-BPs in therapeutic schemes against CRC [19]. An important constraint of these drugs is represented by their preferential localization in the bone marrow [28,29]. To enhance N-BPs tumor targeting, besides the nanoformulated ZA [33], we recently proposed a novel antibody-drug conjugate (ADC), made of ZA and the therapeutic anti-epithelial growth factor receptor (EGFR) antibody cetuximab (Cet) [34]. CRC organoids primed with Cet-ZA ADC could trigger the expansion of Vδ2 T cells from peripheral blood and tumor-infiltrating lymphocytes, able to exert autologous tumor cell killing [34].

In this work, we show that CRC-TAF exposed to soluble ZA or Cet-ZA ADC become able to trigger the proliferation of Vδ2 T cells, starting from purified T lymphocytes. The expanded Vδ2 T cell population was in part represented by effector memory cells that exerted relevant cytotoxic activity towards CRC cells and CRC-TAF when primed with ZA or Cet-ZA. These findings indicate that the Cet-ZA ADC formulation can lead to the precise delivery of ZA to EGFR^+^ cells, targeting TAF besides CRC cells and inducing a pro-stimulatory instead of an inhibitory function.

## 2. Materials and Methods

### 2.1. TAF Isolation from Tumor Specimens

CRC specimens were obtained from 20 patients (see Appendix A) during therapeutic surgery at the Surgical Oncology Unit, IRCCS Ospedale Policlinico San Martino, Genoa, upon signed informed consent (Institutional and Regional Ethic Committee approval, PR163REG2014, renewed in 2017). This cohort of CRC patients was composed of newly-diagnosed patients without any kind of therapeutic treatment before surgery. Samples were anonymized as reported and referred in the paper with the year and serial number for brevity [33]; all these CRC-TAF have been used within the 8–9th culture passage.

Primary TAF lines were obtained by mincing CRC mucosa, enzyme digestion and debris removal by soft centrifugation as reported [35]; cell suspensions were then purified by density gradient centrifugation (Lympholyte, Cederlane, Burlington, ON, Canada) and seeded in culture wells for 36 h. Non-adherent cells were removed and adherent cells showing a fibroblast-like morphology were cultured for an additional 7 d in MEMalpha (Euroclone, Milan, Italy) supplemented with 10% foetal calf serum (FCS, Sigma Chemicals Co., St. Louis, MO, USA). At confluence, cell cultures were split and expanded as described [19,31]. Phenotype for Intercellular Adhesion Molecule 1 (ICAM1), Fibroblast Activation Protein (FAP), Thy-1 Cell Surface Antigen (CD90), Endoglin (CD105), Epidermal Growth Factor Receptor (EGFR) and Epithelial Cell Adhesion Molecule (EPCAM) markers were analyzed by indirect immunofluorescence at different time points, along a culture period of two months as described in Section 2.5. TAF were also tested for the expression of BTN3A1 and BTN2A1.

### 2.2. ZA and Cet-ZA ADC

Zoledronic acid (ZA, MW 272.09) was purchased from Selleckchem (Houston, TX, USA), while Cet (Erbitux^®^) was obtained as a left-over of the preparation used for CRC patients’ therapy (kind gift from the Pharmacy Unit of IRCCS Ospedale Policlinico San Martino, Genoa). Cet-ZA ADC was prepared by Nanovex Biotechnologies (Asturias, Spain) as reported in [34]. Briefly, Erbitux^®^ was dialyzed to eliminate the excipients, while ZA and 1-ethyl-3-(3-dimethylaminopropyl) carbodiimide hydrochloride (EDC, Sigma Aldrich, St. Louis, MO, USA) were prepared in 0.1 M imidazole at pH 6 ± 0.2 (with HCl 1 N). ZA has been incorporated into Cet structure exploiting the phosphoric groups of ZA according to the reactions reported to conjugate peptides and the free phosphoric acid of DNA. Cet-ZA was then purified from free imidazole, EDC and ZA through dialysis with the Slide-A-Lyzer Cassette (Thermo Fisher Italia, Monza, Italy) at 10,000 MWCO for 24 h. The resulting link between ZA and Cet is covalent in the absence of a linker molecule. The drug antibody ratio was 4.3 as determined by matrix assisted laser desorption ionization mass spectrometry [34].

### 2.3. CRC-TAF/T Cell Co-Cultures

Peripheral blood mononuclear cells (PBMC) were isolated from venous blood samples of healthy donors of the Transfusional Centre of the IRCCS Ospedale Policlinico San Martino (provided signed institutional informed consent, defined by the Regional Ethic Committee, CONSAZH780148/17 July 2015) by density gradient separation [36]. Monocytes (Mo) and T lymphocytes were purified from PBMC with the specific negative separation kits (Rosettesep, StemCell Technologies, Vancouver, BC, Canada), with a yield of >75% pure for CD14^+^ Mo or >98% of CD2^+^CD3^+^ T cells (Appendix A). The CD2^+^CD3^+^ T cells were incubated overnight in RPMI 1640 (supplemented with 10% FBS, Penicillin/Streptomycin and L-Glutamine, all from Gibco) to discard by adherence any residual Mo component and to avoid their presence in stimulation experiments. Purified T lymphocytes were added to either TAF or Mo, previously seeded overnight into 96 w U-bottomed plates (Sarstedt, Nümbrecht, Germany), at T:TAF ratios as indicated (from 20:1 to 1000:1), or at a T:Mo ratio of 200:1, according to our published data [33], without or with 2.5 µg/mL Cet-ZA or 1.2 µM ZA. Titration experiments using ZA at 2.5 to 0.6 µM were also performed as described in [33]. After 24 h, at 37 °C in a humidified 5% CO_2_ incubator, human recombinant interleukin 2 (IL-2, Peprotech-Thermo Fisher, 30 IU/10 ng/mL final concentration, Waltham, MA, USA) was added and co-cultures were continued up to 21 days. The concentration of Cet-ZA for functional experiments was determined on the basis of the EC_50_ as reported [34].

### 2.4. Vδ2 T Cell Proliferation

To measure the proliferation of Vδ2 T lymphocytes, purified T cells were labelled with CFSE as described in [31]. Briefly, 10^6^ cells were incubated for 30 min at 37 °C in a water bath, in a complete medium with 100 nM CFSE. Then, cells were extensively washed and put in co-cultures as described in Section 2.3, with or without ZA (1.2 μM or as reported in Section 2.3 for T:TAF and T:Mo co-cultures) or Cet-ZA (2.5 μg/mL). At 10 or 20 d, the percentage of Vδ2 T lymphocytes was evaluated by flow cytometry after labelling with the anti-Vδ2 specific mAb γδ123R3 (IgG1) [33] followed by Alexafluor 647 goat anti-mouse (GAM). Samples were analyzed on a Cytoflex S flow cytometer and the proliferation was indicated by the reduction of CFSE in the cell generations (G) compared to the content of CFSE in the parental (P) component as evaluated by the FlowJo^TM^ software v 10 (Becton Dickinson, Ashland, OH, USA).

### 2.5. Immunofluorescence and Flow Cytometry

For Vδ2 T lymphocytes identification, the anti-T cell receptor (TCR) Vδ2 γδ123R3 (IgG1) and the anti-CD3 JT3A 289/11/F10 (IgG2a) mAb were used, followed by PE-GAM IgG1 or APC-GAM IgG2a (Thermo Fisher Scientific, Waltham, MA, USA). Vδ2 T cells present in cell culture were also characterized by polychromatic immunofluorescence using anti-Vδ2, anti-CD27 (LT27, IgG2a; Bio-Rad, Hercules, CA, USA) and anti-CD45RA (HI100, IgG2b; Bio-Rad) [33] mAb followed by anti-isotype-specific alexafluor 488 or PE or alexafluor 647 conjugated GAM, respectively. Immunofluorescence on CRC TAF was performed with the anti-ICAM1 (14D2D12, IgG1) [31] mAb, anti-CD90 (FAB2067p, IgG2a, R&D Bio-techne, Minneapolis, MN, USA) mAb, the anti-CD105 mAb (from the producing hybridoma purchased from the American Type Culture Collection, ATCC, Manassas, VA, USA), the anti-EPCAM mAb (ab98003, IgG1, Abcam, MA, USA), the anti-FAP mAb (F11-24, IgG1, eBioscience, San Diego, CA, USA), followed by anti-isotype GAM antisera (Southern Biotechnology, CA, USA) conjugated with alexafluor 647 (Invitrogen, Thermo Fisher Scientific). Control samples were stained with fluorophore-conjugated isotype control mAbs or isotype-matched irrelevant mAb (Becton Dickinson, Palo Alto, CA, USA) followed by anti-isotype-specific GAM-alexafluor 647. The expression of butyrophilins was checked using the rabbit anti-BTN2A1 (1:300, Bioss Antibodies, MA, USA) or the rabbit anti-BTN3A1 (1:200, NovusBio. R&D Bio-techne) followed by a fluorescein-conjugated anti-rabbit antiserum (Thermofisher, Waltham, MA, USA). For cytoplasmic staining of BTN3A1 and BTN2A1, cells were first fixed with 4% paraformaldehyde, washed and permeabilized with 1% Triton-X100. The anti-EGFR Cet or the Cet-ZA ADC were incubated for 1 h at RT, followed by the APC-labelled anti-human Ig (APC-α-hIg) antiserum. Controls aliquots were stained with APC-α-hIg antiserum. Samples were run on Cytoflex S flow cytometer (Beckman-Coulter, Pasadena, CA, USA), analyzed with the CytExpert 2.4 computer program (Beckman Coulter) and results expressed as Log fluorescence intensity vs. number of cells (MFI, arbitrary units, a.u.) or percentage of positive cells [33].

### 2.6. Confocal Microscopy

For confocal microscopy, CRC-TAF were plated in 96 w clear flat-bottomed black well plates for imaging (Eppendorf, Merck KGaA, Darmstadt, Germany) and incubated with 2.5 µg/mL/10^6^ cells Cet-ZA ADC for 1 h at RT, followed by the APC-α-hIg antiserum and 100 nM Syto16 counterstaining (Thermo Fisher Scientific). To show Cet-ZA internalization, samples were incubated with the ADC as above at 37 °C for 48 h, fixed with 1% PFA, permeabilized with 1% Triton X-100, followed by FITC-α-hIg antiserum, anti-LAMP-1 mAb (clone H4A3, Thermo Fisher Scientific), followed by GAM anti-isotype-specific alexafluor647 antiserum and Syto Orange (40 nM, Thermo Fisher). Samples were observed by the FV500 confocal Laser Scanning Microscope System equipped with Argon, He-Ne green and He-Ne red lasers, associated with IX81 motorized microscope (Olympus Europe GmbH, Hamburg, Germany), using a PlanApo 20X NA1.00 objective. Each image was taken in sequence mode, to avoid the cross-activation of fluorochromes, and data were analyzed with FluoView computer software v5.0 (Olympus). Results are shown in pseudocolor as red or green fluorescence vs. nuclei in blue [34].

### 2.7. Cytotoxicity Assay and Interferon γ (IFNγ) Production

Cytolytic activity of Vδ2 T cells against CRC-TAF or CRC cell lines was tested with the crystal-violet assay as described previously [32]. Target cells were: CRC-TAF 15-066, 16-001, 16-004, 16-027, 16-030, 16-035, or the CRC cell lines HT29, DLD1 (EGFR^+^) and SW620 (EGFR^dull^) obtained from the cell bank of the IRCCS Ospedale Policlinico San Martino (kind gift of Biological Resource Cell Bank unit, Dr. Barbara Parodi) and correctly identified by the STR DNA profile. The E:T ratio chosen was 10:1 since at this ratio Vδ2 T cells exert low spontaneous anti-tumor cytotoxicity allowing the detection of ZA and Cet-ZA-mediated effect [34]. Experiments were then carried out without or with 2.5 µg/mL Cet-ZA, or 2.5 µg/mL Cet, or 1.2 µM ZA. Indeed, Vδ2 T cell populations used in these experiments were selected for the expression of CD16/FcγRIIIa on more than 60% of cells as detected by immunofluorescence. To detect cytotoxicity, after 48 h of incubation surviving, cells were stained with crystal-violet and colorimetric intensity was evaluated at the wavelength of 594 nm with the fluorimeter VICTORX5 (Perkin Elmer Italia, Milan, Italy). These OD values were compared to the ones obtained in control CRC cells or TAF cultured alone, as a standard of a 100% viability. A representative example of this procedure for the evaluation of TAF viability is shown in Appendix A.

IFNγ was measured in the supernatant of Vδ2 T cells either alone or co-cultured with CRC-TAF without or with 1.25 µM ZA or 2.5 µg/mL Cet-ZA, by ELISA with the specific kit (Peprotech-Thermofisher). Experiments were carried out at the Vδ2 T:TAF ratio of 10:1 (2 × 10^5^ Vδ2 T cells, 2 × 10^4^ TAF) in 24 well-plates for 48 h, as described [31]. The amount of the IFNγ was expressed as pg/mL referred to a standard curve.

### 2.8. Immunohistochemistry (IHC) and Digital Image Analysis

Sections of 4-µm-thick CRC samples were cut and stained with the rabbit anti-BTN2A1 (1:300, Bioss Antibodies) and rabbit anti-BTN3A1 (1:200, NovusBio). IHC was carried out using the automated stainer BOND RX (Leica Biosystem Italia Milan, Italy) according to the manufacturer’s instruction as previously described in detail [34]. Digital images were captured using the Aperio AT2 scanner (Leica) under 20X or 40X objective magnification, stored in Aperio E-slide manager, and analyzed using the Aperio Scan Scope software v.102.0.7.5 (Leica Biosystems, Aperio Technologies, Nussloch GmbH, Nußloch, Germany) [34]. Genie Classifier AI software v.1 (Leica Biosystem) was used to identify and quantify the percentage area of each compartment (tumor vs. stroma), excluding empty or adipose tissue areas. Tumor areas were defined by contouring as described [34], considering the central part of the tumor only. The software was trained, using a panel of images manually annotated by the operator, to distinguish the regions of interest (ROI). According to the manufacturer, >95% training accuracy was required for each classifier in order to proceed with the image analysis of cohort samples. The percentage of tumor and stroma was normalized against the total tissue area (mm^2^) in each image. Two pathologists checked that stromal areas were identified correctly [34,37].

### 2.9. Statistical Analysis

Data are presented as mean ± SD as indicated. Statistical analysis was performed using two-tailed unpaired Student’s *t* test, with Welch correction, using the GraphPad Prism software 5.0. The cut-off value of significance is indicated in each figure legend.

## 3. Results

### 3.1. CRC-TAF Can Stimulate the Expansion of Vδ2 T Cells with Effector Phenotype

First, primary TAF obtained from CRC specimens of 20 patients (Appendix A) were analyzed for the expression of different molecules, including mesenchymal or epithelial markers, adhesion molecules, growth factors or ligands for regulating receptors. As shown in Appendix A (panel A: two representative cases; panel B: 20 cases), CRC-TAF expressed the fibroblast markers FAP, endoglin (CD105) and the adhesion molecule ICAM1, with a low expression of Thy-1 (CD90). All the CRC-TAF reported in this study expressed EGFR on their surface, while the epithelial marker EPCAM was negative (Appendix A, two representative cases in panel A, 20 cases in panel C). Moreover, they were vimentin and transglutaminase II-positive as described previously both for cultured and in situ fibroblasts [19,20]. This phenotype was checked during the culture and remained stable at least for two months (not shown).

To verify the stimulating effect of ZA, purified T lymphocytes from healthy donors were co-cultured with CRC-TAF at different T:TAF ratios in the presence of 2.5 µM ZA, concentration in the range 1–5 µM reported as efficient in our previous papers [19,33], followed by the addition of IL-2. The percentage of Vδ2 T lymphocytes, evaluated at day 10 by flow cytometry with the anti-Vδ2 specific mAb γδ123R3, raised in the presence of ZA at the T:TAF ratios of 200:1 or 1000:1 (Figure 1A) suggesting that the drug can be effective on CRC-TAF. The effect of ZA was analyzed for comparison, in co-cultures of T lymphocytes and monocytes since the latter cells are known to accumulate PA in response to N-BPs leading to Vδ2 T lymphocyte proliferation [25,33].

Subsequent titration experiments were performed to define the EC_50_ of ZA in this system evaluated at day 20 to allow further expansion of Vδ2 T cells. T cells were co-cultured with CRC-TAF or with Mo, in the absence or presence of ZA (from 2.5 µM to 0.6 µM). Vδ2 T lymphocytes increased in all T:TAF co-cultures, with an EC_50_ of 1.2 µM in keeping with our previous data [33]. This confirms that CRC-TAF can be exploited as stimulating cells, although with an efficacy lower than that of Mo (Figure 1B). Kinetics experiments showed that the optimal effect of 1.2 µM ZA was detectable at day 20 (Figure 1C), both for CRC-TAF (range 35–75% Vδ2 T cells) and Mo (>95% Vδ2 T cells). To confirm that Vδ2 T cells detected in the co-cultures were proliferating, another series of experiments was performed using CFSE labeling. Figure 1D depicts an example of these experiments, where CFSE-labelled T cells from donor #42 were cultured alone or co-cultured with CRC-TAF 16-030, without or with 1.2 μM ZA. At day 20, proliferating Vδ2 T cells identified by the decreased level of CFSE content and reacting with the anti-Vδ2 specific mAb were about 45% (Figure 1D, the upper right plot, upper left quadrant) when co-cultures were primed with ZA compared with <1% without ZA (Figure 1D, central right plot, upper left quadrant). The computerized analysis of gated Vδ2 T cells with decreasing content of CFSE showed that the majority of them was in G1-G3 generation (Figure 1E,F).

We then analyzed the phenotype of the Vδ2 T cell populations obtained after 20 d of culture, focusing on the subsets of naive (N) cells that bear both CD27 and CD45RA molecules, central memory (CM) showing only CD27, effector memory (EM) that are double negative and terminal-differentiated memory cells (TEMRA) that are positive for only CD45RA [19,38,39]. As shown in Figure 2A,B (polychromatic immunofluorescence assay and flow cytometry analysis on Vδ2 gated cells, see Appendix A), the majority of Vδ2 T lymphocytes expanded in co-culture with CRC-TAF exposed to ZA, displayed the phenotype of EM cells (>45% CD45RA^−^ CD27^−^) and of TEMRA (CD45RA^+^CD27^−^). Indeed, at the beginning of the culture, the large majority of Vδ2 T cells were naïve cells and they were converted into TEMRA by the addition of IL2 or IL2 and TAF without ZA (Figure 2B, central graph); whereas EM cells represented a large portion of Vδ2 T cells when triggered with ZA and IL2 (Figure 2B, left).

According to its phenotype, this Vδ2 T cell population could kill the three CRC cell lines tested (HT29, DLD1 and SW620) in the presence of ZA. Moreover, cytotoxic activity was inhibited by the anti-TCR Vδ2 mAb γδ123R3 (Figure 2C–E), demonstrating that γδ TCR is involved in target cell recognition. Furthermore, Vδ2 T lymphocytes activated by co-culture with ZA-treated TAF (Vδ2 T_AF_), or with Mo exposed to ZA (Vδ2_Mo_), could kill CRC-TAF at the E:T ratio of 10:1. Moreover, this killing was inhibited by the anti-Vδ2 TCR mAb (Figure 2F,G, respectively) reinforcing the role of TCR in target cell recognition. Altogether, these findings indicate that ZA induces CRC-TAF to stimulate the expansion of anti-tumor effector Vδ2 T lymphocytes that can also exert lytic activity against CRC-TAF.

### 3.2. Cet-ZA ADC Reacts with CRC-TAF and Induces Vδ2 Cytotoxic Effector T Cells

ZA conjugation with Cet was achieved following the scheme described for the chemical reactions of nucleic acids and proteins, exploiting the synthesis of phosporamidate; the reactions, leading to a covalent chemical bond between Cet and ZA, have been recently reported [34]. Cet-ZA ADC reactivity was verified by indirect immunofluorescence and flow cytometry using three CRC-TAF (16-001, 16-004 and 16-035) incubated with serial dilutions of the ADC, followed by APC-labelled anti-human Ig antiserum. Figure 3A shows that the reactivity of the ADC (flow cytometry profiles in panel 3B) was comparable to that of native Cet, optimal at 2.5 µg/mL/10^6^ cells, this concentration was chosen for the functional experiments. Confocal microscopy, depicted in Figure 3C, showed that Cet-ZA ADC was internalized in CRC-TAF and colocalized with the endosomal marker LAMP-1 within 48 h. Appendix A represents confocal images of Cet-ZA reactivity and cell distribution on three representative CRC-TAF cell lines compared to native Cet.

We next verified whether Cet-ZA ADC can deliver ZA into CRC-TAF, through the binding with EGFR, and trigger the activation of Vδ2 T cells. To this aim, T lymphocytes purified from healthy donors were co-cultured with CRC-TAF and serial dilution of Cet-ZA (5 µg/mL, 2.5 µg/mL or 1.25 µg/mL) and IL-2. On day 10 of co-culture in the presence of Cet-ZA, an evident increase of the percentage of Vδ2 T cells was detectable. Indeed, the percentage of Vδ2 T cells ranged from 27 to 58 at 5µg/mL of Cet-ZA and the EC_50_ of the ADC was about 2.5 µg/mL (Figure 4A, 25% Vδ2 T cells vs. <5% without Cet-ZA). Subsequent experiments performed with CRC-TAF and Cet-ZA at 2.5 µg/mL for 10 or 20 days, showed that the latter time point is required to reach the percentage of Vδ2 T lymphocytes ranging between 30 and 58% (Figure 4B, mean 40%), starting from 1–4% of the initial T cell population.

To analyze the ability of Cet-ZA to elicit Vδ2 T cell-mediated anti-tumor activity, the Vδ2 T cell populations obtained from healthy donors as described in Section 3.1 were used as effectors in cytotoxicity assay. Five populations enriched at 60% of Vδ2 T lymphocytes were chosen and challenged with the CRC cell lines HT29 and DLD1 (EGFR^+^), or SW620 (EGFR^dull^). Figure 4 shows that in the presence of Cet-ZA the percentage of viable HT29 and DLD1 cells (Figure 4C,D) significantly decreased to 30% in co-cultures with Vδ2 T lymphocytes. This effect was stronger than that obtained with native Cet (40–50% of viable CRC cells, not shown). The EGFR^dull^ SW620 cell line was less sensitive to the effect of Cet-ZA (Figure 4E), nevertheless it was still responsive to soluble ZA (Figure 2C). Of note, the anti-Vδ2 TCR mAb and the combination of anti-Vδ2 plus anti-CD16 mAbs could significantly inhibit Cet-ZA-induced cytotoxicity (Figure 4C,D) indicating that the ADC can activate tumor cell killing by both TCR and FcγRIIIA. In another series of experiments, Vδ2 T lymphocytes activated with ZA-treated TAF (Vδ2 T_AF_) were employed as effectors against CRC-TAF as above. As shown in Figure 4F, Cet-ZA elicited a significant anti-TAF killing (viability decreased to 20–60%) that was higher than that elicited by native Cet (80% TAF viability) and was inhibited by the anti-Vδ2 TCR mAb, as occurred for soluble ZA (see Figure 2). Activated Vδ2 T cells obtained on ZA-activated monocytes (Vδ2_Mo_) are shown in Figure 4G for comparison. Furthermore, in this case, Cet-ZA was more effective than native Cet in triggering anti-TAF lytic activity (viability decreased to 5–30% vs. 60% of native Cet) that was significantly reduced by the anti-Vδ2 mAb. Interestingly, both ZA and Cet-ZA elicited the release of an antitumor cytokine such as IFNγ [39] in the supernatant of Vδ2:TAF co-cultures (Figure 4H); this finding supports that Cet-ZA ADC can stimulate and amplify the antitumor response mediated by Vδ2 T lymphocytes.

### 3.3. CRC-TAF Express In Situ BTN3A Molecules in Stromal Areas within the Tumor

To support the idea that targeting TAF with ZA or Cet-ZA can be exploited for anti-CRC treatment, the stromal areas of CRC specimens were analyzed by IHC for butyrophilin expression. BTN3A1 or BTN2A1 expression was examined by digital pathology, as reported [34]. BTN2A1 was significantly expressed on cells with fibroblast morphology (Figure 5A,B 17-050 representative case), while BTN3A1 was barely detectable by IHC (Figure 5C, 15-073 representative case). The percentage area of each compartment (tumor vs. stroma) was determined in the CRC specimens with the pattern recognition tool of the Genie Classifier software v.1. Data refer to the areas defined in each slide by contouring the central part of the tumor (CT) as previously described [34]. Results in Figure 5D indicate that stromal areas are significantly higher than tumor zones, thus highlighting the relevance of TME. Of note, the cultured CRC-TAF cell lines 16-004, 16-016, 16-035 expressed BTN2A1 on their surface and in the cytoplasm, evidenced by immunofluorescence and flow cytometry (Figure 5E,F), whereas BTN3A1 membrane and cytoplasmic expression was less detectable (Figure 5E,F). These data point to the potential importance of butyrophilin expression in CRC stromal areas that represent a relevant fraction of the whole tumor tissue.

## 4. Discussion

Two important problems of cancer immunotherapy are the overcoming of a suppressive TME and the specific targeting of tumor cells. In this paper, we show that we can use a new ADC, made of the N-BP ZA and the anti-EGFR cetuximab [34], to bypass both these limits. Here, we demonstrate that Cet-ZA ADC can react with CRC-TAF expressing EGFR, making them able to stimulate Vδ2 T lymphocytes with effector phenotype and anti-tumor function. On the other side, these effectors are able to kill also CRC-TAF, thus limiting the potential suppressive activity of TME.

We recently published the synthesis, the reactivity and the function of this Cet-ZA ADC in a 3D model of human CRC organoids, showing that it is internalized and localized in the endocytic compartment in tumor cells, as occurs in general for ADC whose processing follows the route of the antibody [34,40]. The advantage of this type of ADC, based on a pH-cleavable phosphoramidate linker for the conjugation, is the delivery of the payload inside the cell, where it can exert its function [41]. This is relevant for ZA, as it acts on the mevalonate pathway inducing the production of the PA isopenthenyl pyrophosphate by cells, either antigen-presenting or tumor cells, that in turn stimulate Vδ2 T lymphocytes [24,25]. Accordingly, Cet-ZA is internalized in CRC-TAF colocalizing with the endocytic marker LAMP-1, although the kinetics of internalization in TAF is slower (24–48 h) than in CRC cells (1–2 h) [34].

CRC-TAF are equipped to allow ZA, either soluble or driven by the ADC, to exert its effect: they express BTN2A1, and to a lesser extent BTN3A1 molecules that are needed for the exposure of PA in a molecular form recognizable by the Vδ2 TCR [26,27]. Thus, CRC-TAF exposed either to soluble ZA or to Cet-ZA ADC, can elicit the proliferation of Vδ2 T lymphocytes displaying the phenotype of effector-memory cells, that is cells mostly CD45RA^−^CD27^−^. Interestingly, PA accumulation occurring in antigen presenting cells and in cancer cells following N-BPs exposure, has been reported to drive not only proliferation, but also maturation of Vδ2 T lymphocytes from naïve to memory cells [39,42]. These lymphocytes are able to kill both tumor cells and CRC-TAF themselves, thus resulting in a regulatory feedback, besides a direct anti-cancer function.

In particular, the cytotoxicity triggered by Cet-ZA ADC is both TCR-mediated, due to the PA production consequent to ZA effect, and antibody-mediated (ADCC), due to Cet binding to FcγRIIIA/CD16. The latter type of killing is typical of several different immune effectors, including natural killer cells and macrophages [20,43], and represents a mechanism widely exploited for the pharmacological function of therapeutic antibodies [43,44]. Moreover, the double action of this ADC provides a dual gun to kill tumor cells, on one side, and down-regulate the function of CRC-TAF on the other side. We have already reported that N-BPs, acting on lymph-node mesenchymal stromal cells, can induce Vδ2 T helper1/effector memory lymphocytes and rescue the recognition and killing of lymphoma cells, facilitating rituximab-induced ADCC [39]. Cet-ZA ADC brings together the ability to stimulate effector/memory cells, elicit antitumor activity and allow the action of the therapeutic mAb. At the tumor site, where TAF, Vδ2 T lymphocytes and cancer cells are close one to each other, Cet-ZA can turn TAF into stimulators of effectors that first eliminate stromal cells that may interfere with anti-cancer immune response, and then become effective cancer killers.

ZA/Cet-ZA effect on Vδ2 T cell proliferation is not evident at the lowest T:TAF ratios conceivably due to the reported inhibiting action exerted by fibroblasts or mesenchymal stromal cells on leukocyte function [4]. Nevertheless, Cet-ZA-elicited cytotoxicity is detectable also at low E:T ratio according to what observed for soluble ZA in the present work and previous reports [39]. We can hypothesize that effector lymphocytes proliferated in areas poor of stromal cells can exert antitumor activity also in areas enriched with TAF.

We have recently described ZA-encapsulated nanoparticles (ZA-SPN) as a drug formulation suitable to direct ZA preferentially into the tumor site avoiding the bone tropism of N-BPs [33]. A major advantage of the Cet-ZA ADC towards other drug formulations, including ZA-SPN, is the ability to select, activate and drive an anti-tumor effector cell population into the tumor, due to the specificity of the anti-EGFR Cet, not only avoiding other localizations of ZA, but also targeting only EGFR^+^ cells, either CRC cells or CRC-TAF. A possible drawback is represented by the relative unresponsiveness of CRC cells expressing low levels of EGFR that might allow their escape from the targeting with Cet-ZA. Nevertheless, ZA is delivered precisely inside the cell that has to become the target of immune effectors; afterward, the function of the drug will be enabled provided the expression of the butyrophilins needed to express the PA at the cell surface.

It is of note, that not only cultured CRC-TAF, but also in CRC tissue histology, stromal cells express BTN2A1. BTN3A1 is barely detectable in situ and mainly in the cytoplasm of cultured CRC-TAF, possibly needing an undefined signal (e.g., an inflammatory cytokine) to be expressed by tumor stroma. On the other hand, we reported that BTN3A1 is the main butyrophilin expressed by tumor epithelial cells in CRC [34]. Thus, fibroblasts and tumor cells apparently bear on their membrane different molecules that may cooperate to allow the complete pharmacological action of ZA. This is of interest in the perspective of novel anticancer therapies [45], considering that stroma frequently represents a sizable fraction of tumor tissue and fibroblasts are its principal population. From this viewpoint, a recent report evidenced that TAF produce EGF upon Cet treatment, supporting CRC cell growth and rescuing tumor cells from immunotherapy [46]; thus, CRC-TAF should be considered as targets of therapy to overcome this TME-mediated drug resistance.

## 5. Conclusions

In conclusion, double targeting of CRC and CRC-TAF by Cet-ZA ADC may represent an interesting therapeutic tool to enhance anti-tumor immunity, restrict the response to the tumor site and convert potential suppressive cells into cooperative anti-cancer agents.

## 6. Patents

The Cet-ZA ADC is under the Italian patent n. 102022000003758 on 1 March 2022.

## Figures and Tables

**Figure 1 cancers-15-00610-f001:**
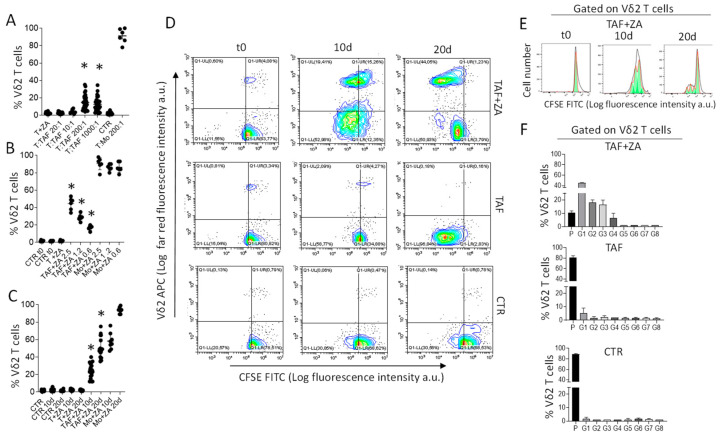
ZA-primed CRC-TAF trigger Vδ2 T cell proliferation. Vδ2 T lymphocytes were identified at day 10 of culture (**A**) or day 10 and day 20 (**B**,**C**) by flow cytometry, and results are expressed as percentage of Vδ2 T cells/total T cells. The mean ± SD from 20 cases. (**A**): T lymphocytes from healthy donors (*n* = 10, #19-#28) co-cultured with CRC-TAF (*n* = 20 patients) at the indicated T:TAF ratios without (CTR) or with 2.5 µM ZA and IL-2. (**B**): co-cultures of T cells (*n* = 2 #35 and #36) and CRC-TAF (*n* = 6, 016-001, 16-004, 16-030, 16-035, 16-039, 16-047) at T:TAF ratio 200:1 or Mo from healthy donors (*n* = 3, #20, #23, #24) at T:Mo ratio 200:1, without (CTR) or with ZA (from 2.5 µM to 0.6 µM). (**C**): co-cultures of T cells (*n* = 10, #35, #37, #38, #40, #41, #42, #50, #52 #55, #58) and CRC-TAF (*n* = 20, T:TAF ratio 200:1) or Mo (T:Mo ratio 200:1) from healthy donors (*n* = 6), in the absence (CTR) or presence of ZA (1.2 µM). T + ZA in panels A-C: percentage Vδ2 T cells in cultures of T lymphocytes with 1.25 µM ZA. (**D**): Vδ2 T cell proliferation upon a co-culture of T cells from donor #42 labeled with 100 nM CFSE, washed and cultured alone (upper plots) or co-cultured with CRC-TAF 16-030, without (**central** dot plots) or with 1.2 μM ZA (lower dot plots). At time 0 (**left** plots) or at 10 d (**central** plots) and 20 d (**right** plots), the percentage of Vδ2 T lymphocytes was evaluated. Percentages of proliferating Vδ2 T cells in the total cell culture are reported in the upper **left** quadrants of each plot. (**E**): Cell proliferation analysis of a representative case in the co-culture with T and TAF + ZA experimental condition. (**F**): percentage of Vδ2 T cells in the parental (P) or subsequent generations (G1–G10) in the culture conditions indicated. In all panels * *p* < 0.001 vs. CTR.

**Figure 2 cancers-15-00610-f002:**
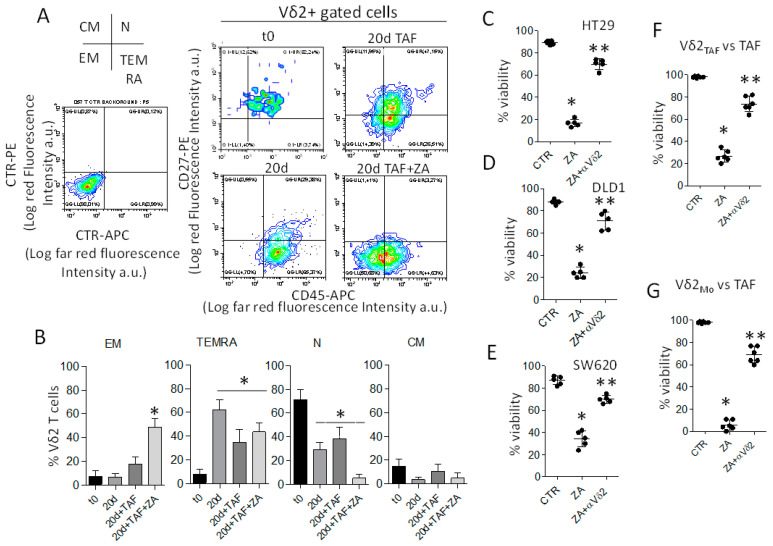
ZA induces CRC-TAF to stimulate the expansion of anti-tumor effector Vδ2 T lymphocytes. (**A**): Phenotype of Vδ2 T cells (one representative experiment) expanded on untreated or ZA-treated (1.2 µM) CRC-TAF (16-030) on day 20. Samples were labelled with anti-Vδ2, anti-CD27 and anti-CD45RA mAbs gated on Vδ2 T cells and results are expressed as Log far red fluorescence intensity vs. Log red fluorescence intensity. The cross depicted on the **left** identifies the quadrants showing effector memory (EM, lower **left**, double negative), terminally-differentiated effector memory (TEMRA, lower **right** CD45RA^+^ CD27^−^), naive (N, upper **right**, double positive) and central memory (CM, upper **left**, CD45RA^−^ and CD27^+^) cells. (**B**): Analysis of the phenotype of Vδ2 T cells obtained as in A, using T lymphocytes from three healthy donors (#45, #46, #47) co-cultured with six CRC-TAF (016-001, 16-004, 16-030, 16-035, 16-039, 16-047). Results are expressed as percentage of EM, TEMRA, N and CM cells gated on Vδ2 T cells; the mean ± SD from 18 experiments is shown. (**C**–**E**): Activated Vδ2 T cells obtained as in A from five healthy donors were tested in a cytotoxicity assay vs. the CRC cell lines HT29 (**C**) and DLD1 (**D**) (EGFR^+^) or SW620 ((**E**), EGFR^dull^) at the E:T ratio of 10:1, without (CTR) or with soluble ZA (1.2 µM). In some samples, the anti-Vδ2 TCR mAb was added. Results are expressed as percentage of viability evaluated by crystal-violet assay. (**F**,**G**): Activated Vδ2 T lymphocytes from two healthy donors obtained with ZA-activated CRC-TAF (Vδ2 T_AF_) or monocytes (Vδ2_Mo_) were employed as effectors against CRC-TAF (*n* = 3, 016-001, 16-004, 16-030) as targets at the E:T ratio of 10:1, as above. Cytotoxicity was detected as in panels (**C**–**E**), and referred to as percent of viability. * *p* < 0.001 ZA vs. CTR and ** *p* < 0.001 ZA + αVδ2 vs. ZA.

**Figure 3 cancers-15-00610-f003:**
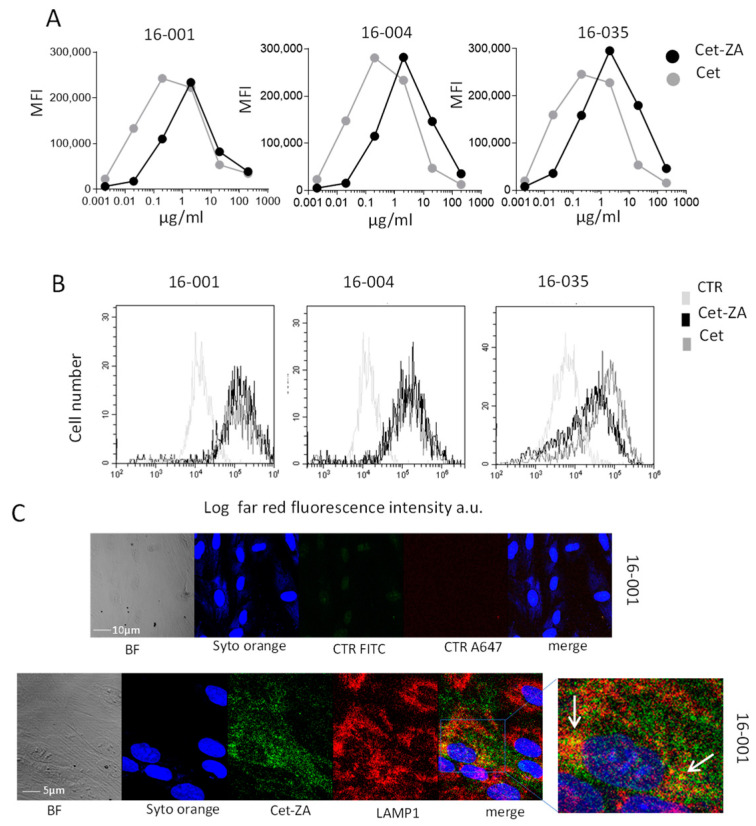
Cet-ZA ADC reactivity with CRC-TAF. (**A**): CRC-TAF (16-001, 16-004 and 16-035), incubated with serial dilutions of Cet-ZA ADC, followed by APC-anti-hIg antiserum were analyzed with the CytExpert 2.4 software and results expressed as mean fluorescence intensity (MFI, a.u.). (**B**): The three CRC-TAF cell lines were incubated with 2.5 µg/mL/10^6^ cells of Cet-ZA ADC (black lines) or native Cet (dark grey lines), followed by APC-anti-hIg antiserum, run on the flow cytometer and analyzed as in A. Results are expressed as Log red fluorescence intensity (MFI, a.u.) vs. number of cells. Cells stained with the second reagent alone (CTR) are shown with light grey lines in each subpanel. (**C**): Upper images show CRC-TAF 16-001 stained with Syto orange alone depicted in blue pseudocolor. Lower images represent Cet-ZA internalization. Blue pseudocolor: nuclei evidenced by Syto Orange; green: Cet-ZA reactivity; red: anti-LAMP-1 mAb reactivity. The enlargement of merge staining is shown on the **right**. White arrows: colocalization of Cet-ZA and LAMP-1 (yellow). BF: bright field. Merge: overlay of blue, green and red fields. Samples were observed by the FV500 confocal Laser Scanning Microscope System using a PlanApo 20X NA1.00 objective. Images were taken in sequence mode to avoid cross-talk between the fluorochromes.

**Figure 4 cancers-15-00610-f004:**
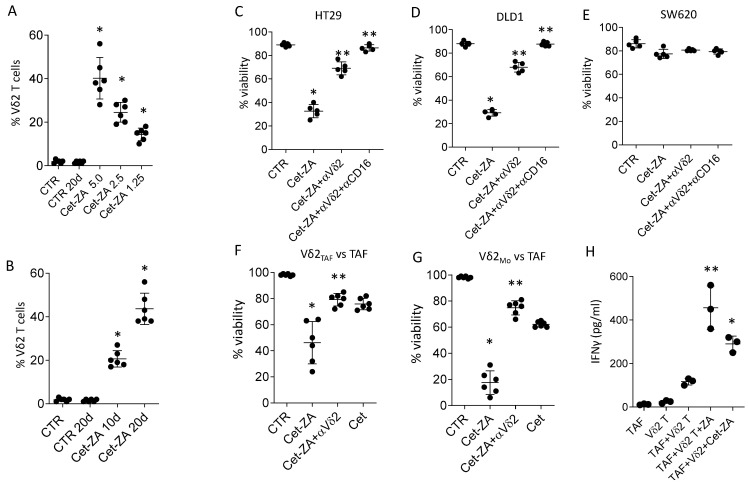
Cet-ZA can expand anti-tumor Vδ2 T cells and stimulate cytolytic activity. (**A**): Purified T lymphocytes from healthy donors (#49, #50) were co-cultured with CRC-TAF 16-001, or 16-004 or 16-035 without (CTR) or with Cet-ZA (5 µg/mL, 2.5 µg/mL or 1.25 µg/mL) and IL-2 and analyzed on day 10 by flow cytometry. Results are expressed as percentage Vδ2 T lymphocytes; the mean ± SD from 6 experiments is also shown. (**B**): Experiments performed as in A without (CTR) or with Cet-ZA at 2.5 µg/mL for 10 or 20 days; results expressed as percentage of Vδ2 T lymphocytes also showing the mean ± SD from 6 experiments from co-cultures of T (#51, #52 and CRC-TAF (16-001 or 16-004 or 16-035). (**C**–**E**): Activated Vδ2 T cells obtained from healthy donors (*n* = 5) were used as effectors in cytotoxicity assay vs. HT29 or DLD1 ((**C**,**D**), EGFR^+^) or SW620 (E, EGFR^dull^) without (CTR) or with Cet-ZA (2.5 µg/mL) or Cet (2.5 µg/mL) or soluble ZA (1.2 µM). In some samples the anti-Vδ2 TCR mAb or the anti-CD16 mAb were added (1 µg/mL). Cytotoxicity was detected by crystal-violet staining and expressed as percent of viability. (**F**,**G**): Activated Vδ2 T lymphocytes obtained with ZA-activated CRC-TAF (Vδ2 T_AF_) or monocytes (Vδ2_Mo_) were employed as effectors against CRC-TAF as above. Cytotoxicity was detected and expressed as in (**C**–**E**). (**H**): IFNγ measured in the supernatant of Vδ2 T cells (#50) either alone or co-cultured with CRC-TAF (16-001 or 16-030 or 16-035), without or with 1.25 µM ZA or 2.5 µg/mL Cet-ZA, by ELISA. Results are expressed as pg/mL referred to a standard curve. In (**A**–**G**) panels: * *p* < 0.001 vs. CTR, ** *p* < 0.001 vs. Cet-ZA. In (**H**) panel: ** *p* < 0.004 vs. TAF + Vδ2 T and * *p* < 0.001 vs. TAF + Vδ2 T.

**Figure 5 cancers-15-00610-f005:**
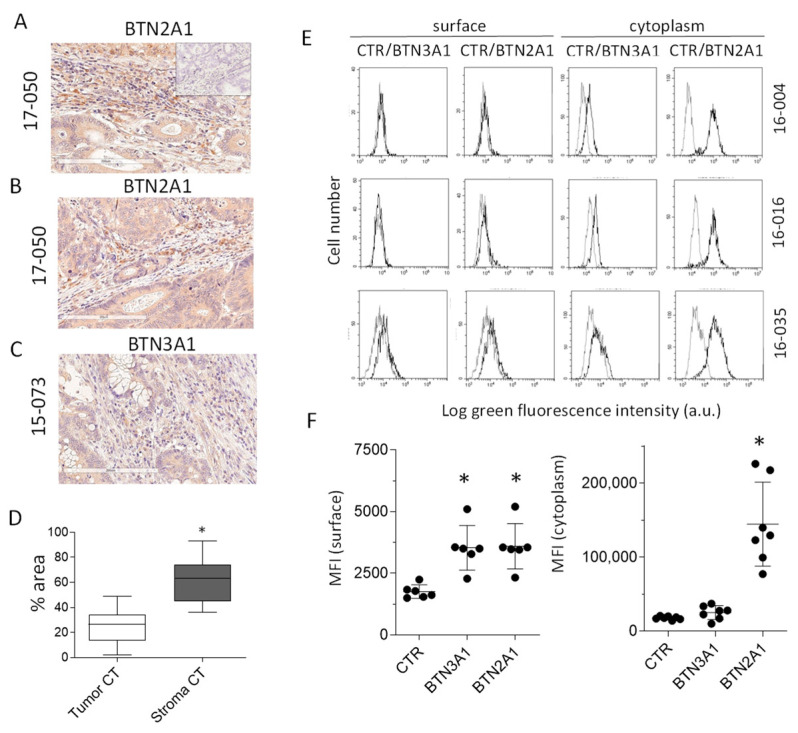
Expression of butyrophilins on tumor stroma and cultured CRC-TAF. (**A**–**C**): Sections of the CRC 17-050 or 15-073 were stained with the rabbit anti-BTN2A1 (two fields of 17-050 in (**A**,**B**)) or the rabbit anti-BTN3A1 (15-073, (**C**)) antisera respectively. Digital images (20×) were captured using the Aperio AT2 scanner. Negative control in the inset of A. (**D**): Samples acquired as in (**A**–**C**) were analyzed with the pattern recognition tool of the Genie Classifier software and the percentage area of each compartment (tumor vs. stroma) was calculated specifically in the central tumor. The results are the mean ± SD from 6 different CRC specimens, each analyzed in 4 slide areas. * *p* < 0.001. (**E**) **Left**: CRC-TAF (16-004,16-016,16-035) were surface stained with the anti-BTN2A1 or the anti-BTN3A1 antisera, followed by FITC-anti-rabbit antiserum. (**E**) **Right**: the same samples were fixed and permeabilized before staining. Samples were analyzed by flow cytometry and results expressed as the Log green fluorescence intensity vs. number of cells (MFI, arbitrary units, a.u.). Grey histograms: negative control (CTR); black histograms: BTN2A1 or BTN3A1. (**F**): MFI of BTN3A1 and BTN2A1 in six CRC-TAF stained at the surface (**left**) or in the cytoplasm (**right**). Mean ± SD of the six cases; * *p* < 0.001.

## Data Availability

The data and reagents will be available under appropriate and motivated requests upon a material transfer agreement between the IRCCS Ospedale Policlinico San Martino and the other party.

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
