# Peer review of "Priming of Colorectal Tumor-Associated Fibroblasts with Zoledronic Acid Conjugated to the Anti-Epidermal Growth Factor Receptor Antibody Cetuximab Elicits Anti-Tumor Vδ2 T Lymphocytes"

_cancers, 2023, doi:10.3390/cancers15030610_

Round 1

Reviewer 1 Report

Priming of colorectal tumour-associated fibroblasts with zoledronic acid 2 conjugated to the anti-Epidermal Growth Factor Receptor antibody cetuximab 3 elicits anti-tumour Vδ2 T lymphocytes. 4

Jordi Leonardo Castrillo Fernandez1, Roberto Benelli1, Delfina Costa1, Alessio Campioli2, Sara Tavella2, Maria 5 Raffaella Zocchi3* and Alessandro Poggi1*

Brief summary:

In this manuscript, the authors describe the effect on a new formulation of ZA that target in a more selective way than the previous formulation, colorectal cancer cells. They also provide insights on the mechanism of action of ZA inducing tumor response by having a double killing action, throught the expansion of Vδ2 T lymphocytes.

The authors are forced to use a lot of different acronyms, however the manuscript is well written and the messages are clear and in most of the experiment well supported by the data shown. Even if it is hard to believe in Cet-ZA clinical application, the general concept is really interesting in a translational approach. It could lead to future optimisation of patient care.

General concept comments:

To my point of view, the manuscript presented could be published if some minor corrections and/or precisions are made:

- As stated in the manuscript, the CRC samples were retrieved during therapeutic surgery, however no information is given on the other treatment that could have been given previously to surgery in a neoadjuvant setting (Table 1). These data are crucial when dealing with TAF and ECM modifications. Please complete the table with this information

- In the submitted version, almost all IF and IHC pictures were not visible and pixelized. Moreover some pictures have to be modified to better support the author’s message (cf Specific comments)

- As not tumor cells or TAF express EGFR, there is a high risk of selection during the treatment with Cet-ZA with limited clinical potential effect and application. As so the authors should discuss this point in this section as a limit of the study. They should put it in balance with the respective proportion of tumor cells and TAF that express EGFR. They could emphasise more on the concept of selective targeting as started in the lines 522-530.

- The fact that there is more stroma than tumor tissue is not enough supported by the data (Fig5A). As stated in the M&M, sections from CRC where stained and then the Genie Classifier AI is used to identify and quantify the percentage area of each compartment. However, for colorectal cancer, there is often non-tumoral and adjacent to tumor tissues on whole section slide. In these areas the adipose tissue and empty areas are excluded by the algorithm, but nothing is said about the exclusion of peri-tumoral tissue. A step of tumor contouring should be added to be able to compare tumor vs stroma distribution inside the tumor area.

Specific comments:

Fig 3C : Please provide better quality pictures for the IF especially if the author want to prove the colocalization of LAMP-1 with the Cet-ZA.

Scheme 3B : Please provide better high magnification pictures so that the reader can see precisely the pseudo color

Line 447 to 460 : Please modify the style as everything is in italic.

Author Response

Brief summary:

In this manuscript, the authors describe the effect on a new formulation of ZA that target in a more selective way than the previous formulation, colorectal cancer cells. They also provide insights on the mechanism of action of ZA inducing tumor response by having a double killing action, throught the expansion of Vδ2 T lymphocytes.

The authors are forced to use a lot of different acronyms, however the manuscript is well written and the messages are clear and in most of the experiment well supported by the data shown. Even if it is hard to believe in Cet-ZA clinical application, the general concept is really interesting in a translational approach. It could lead to future optimisation of patient care.

Reviewer 1 

General concept comments:

To my point of view, the manuscript presented could be published if some minor corrections and/or precisions are made:

- As stated in the manuscript, the CRC samples were retrieved during therapeutic surgery, however no information is given on the other treatment that could have been given previously to surgery in a neoadjuvant setting (Table 1). These data are crucial when dealing with TAF and ECM modifications. Please complete the table with this information

We apologize for not clarifying this point this in the M&M section. Our cohort of CRC patients was composed of newly diagnosed patients without any kind of therapeutic treatment before surgery. This detail has been added to M&M section, lines 101-102. 

- In the submitted version, almost all IF and IHC pictures were not visible and pixelized. Moreover some pictures have to be modified to better support the author’s message (cf Specific comments)

Actually, the immunofluorescence and immunohistochemistry images of the PDF file of the manuscript were worse than those in the WORD File. Indeed, figures were embedded into the WORD file after conversion to emf.file that maintained the high resolution quality of the original images. Nevertheless, we tried to get a better resolution in the revised version of the manuscript. Also, we advise the reviewer to look at the WORD file of the manuscript if possible instead of the PDF file.

- As not tumor cells or TAF express EGFR, there is a high risk of selection during the treatment with Cet-ZA with limited clinical potential effect and application. As so the authors should discuss this point in this section as a limit of the study. They should put it in balance with the respective proportion of tumor cells and TAF that express EGFR. They could emphasise more on the concept of selective targeting as started in the lines 522-530.

The point raised by the reviewer is of interest. Our preparations of TAF expressed EGFR at high intensity at the cell surface, as shown in figure S3 panel C (Figure S2 in the previous version). Indeed, the difference among the negative controls and the EGFR expression is of about six log2 fold looking at the mean values. On the other hand, we tested tumor cell lines expressing high amounts of EGFR (HT29 and DLD-1 cell line) comparing them with the low expression of EGFR of SW620 tumor cells. Nevertheless, in the revised version of the manuscript, we discuss more in detail the possibility of selecting tumor cells with low EGFR expression as a potential limitation of the study (see lines 538-540).

- The fact that there is more stroma than tumor tissue is not enough supported by the data (Fig5A). As stated in the M&M, sections from CRC were stained and then the Genie Classifier AI is used to identify and quantify the percentage area of each compartment. However, for colorectal cancer, there is often non-tumoral and adjacent to tumor tissues on whole section slide. In these areas the adipose tissue and empty areas are excluded by the algorithm, but nothing is said about the exclusion of peri-tumoral tissue. A step of tumor contouring should be added to be able to compare tumor vs stroma distribution inside the tumor area.

Of course, the point raised by the reviewer is of interest. Indeed, TAF can be present not only inside the tumor area but also at the tumor margin. We repeated the analysis considering only the central part of the tumor (CT) defined as described (Suppl. Fig.6A of reference 34). The modality of selection of the tumor areas is better described in that reference. Nevertheless, In the revised version of the manuscript we further specify how the analysis has been performed looking at the tumor area avoiding the inclusion of tumor margins and healthy tissue areas (M&M lines 241-242, results lines 456-457).

Specific comments:

Fig 3C: Please provide better quality pictures for the IF especially if the author want to prove the colocalization of LAMP-1 with the Cet-ZA.

We provide a higher magnification of the colocalization (figure 3C). Indeed, in the WORD file of the original manuscript, the quality of figures was good. Unfortunately, the immunofluorescence and immunohistochemistry images of PDF file of the manuscript were worse than those in the WORD File. We tried to get a better resolution in the revised version of the manuscript. Otherwise, we can provide the reviewer with the original

figures to show the good quality of images.

Scheme 3B: Please provide better high magnification pictures so that the reader can see precisely the pseudo color

As replied to the previous point, the immunofluorescence and immunohistochemistry images of PDF file of the manuscript were worse than those in the WORD File. We tried to get a better resolution in the revised version of the manuscript.

Line 447 to 460: Please modify the style as everything is in italic.

We have modified the style accordingly.

Reviewer 2 Report

In this work, the authors studied how ZA and Cet-ZA ADC stimulate the T cell proliferation via activating fibroblasts in CRC. The study is well designed and has valuable clinical implications. But some questions need to be addressed:

1. The errors in language and grammar are found in many places. Please perform an extensive grammar check on the manuscript or ask for professional help for the writing.

2. For the fibroblasts used in the study, how did the  authors determine if they are tumor-associated or just normal fibroblasts? If any of the protein markers was used to distinguish TAFs from normal fibroblast, please add more background introduction in the Result 3.1 with reference. And since there was no cytokine or any other drugs reported in the fibroblast culture media, the reviewer find it very hard to believe the fibroblast can maintain the TAF phenotypes for 8-9 passages. Is there any evidence to support this?

3. For Fig1A, why did the low T:TAF ratio show no response to ZA? How was the time point determined? It would be better if the author can also show the data on the earlier time points.

4. A control group with just T cell and ZA treatment should be added to all the datasets in Figure 1. If this has been done in the previous study, please mention the results and reference in the result description.

5. A brief introduction of interaction among monocytes, Vδ2 T lymphocytes and ZA should be added in the Results 3.1. Also, according to the description in Methods, the monocytes seem to be attached the surface during the co-culture. Are they polarized?

6. The representative gating strategy for Fig2A should be shown either in the same figure or in a supplementary figure.

7. The original images from which the cytotoxicity results in Fig 2&4 were generated should be shown in the same figure or in a supplementary figure. Additional cytokine release data should be added to support the cytotoxicity results.

Author Response

Reviewer 2

In this work, the authors studied how ZA and Cet-ZA ADC stimulate the T cell proliferation via activating fibroblasts in CRC. The study is well designed and has valuable clinical implications. But some questions need to be addressed:

The errors in language and grammar are found in many places. Please perform an extensive grammar check on the manuscript or ask for professional help for the writing.

We have performed additional grammar checks with two specific software on the manuscript, as suggested by this reviewer. We think that the present version of the manuscript has been improved, although reviewer 1 found the previous version well and clearly written. If the present version were still full of mistakes, we will ask for professional native English-speaking expert help for the writing as proposed.

For the fibroblasts used in the study, how did the authors determine if they are tumor-associated or just normal fibroblasts? If any of the protein markers was used to distinguish TAFs from normal fibroblast, please add more background introduction in the Result 3.1 with reference. And since there was no cytokine or any other drugs reported in the fibroblast culture media, the reviewer find it very hard to believe the fibroblast can maintain the TAF phenotypes for 8-9 passages. Is there any evidence to support this?

The reviewer raised a good question. To our knowledge, there are no markers specific for tumor associated fibroblasts (TAF) that can be used to distinguish them from healthy fibroblasts. On the other hand, the in situ analysis by IHC or IF should consider that several antibodies do not work as efficiently as in flow cytometry. We define TAF as adherent cells isolated from the macroscopically neoplastic region of tumor specimens neither expressing epithelial or endothelial markers nor leukocyte-associated markers. The morphology of cells obtained in culture was fibroblast-like, and the markers expressed are typical of fibroblasts or mesenchymal stromal cells. Thus, they can be considered bona-fide as tumor-associated fibroblasts.

As for the fibroblast phenotype during culture, flow cytometry analysis was performed at every passage and the expression of the markers tested was superimposable at least from the first to the 9th passage. Moreover, they were vimentin and transglutaminase II positive as described previously both for cultured and in situ fibroblasts [ref. 19,20]. TAF culture was performed without cytokine added to the medium, but with supplements such as deoxyribonucleic acids and fetal calf serum. Furthermore, it has been reported that TAF can produce EGF by themselves, so we think that the culture conditions used were adequate to get cells with morphological, phenotypical and functional features superimposable to those of fibroblasts. Some of these considerations have been added to the Results section (lines 262-265). 

  1. For Fig1A, why did the low T:TAF ratio show no response to ZA? How was the time point determined? It would be better if the author can also show the data on the earlier time points.

In figure 1A, at the T:TAF of 20:1 or 10:1 there is no triggering of Vδ2 T cells. This is not unexpected, as it has been demonstrated by us and other researchers that TAF can inhibit lymphocyte cell proliferation triggered by mitogenic stimuli. In the literature, there are several reports claiming that fibroblast-like cells, such as mesenchymal stromal cells, mesenchymal stem cells, tumor-associated fibroblasts and healthy fibroblasts can downregulate immune response. This effect is strictly related to the high number of stromal cells with respect to T cells.   Actually, we found that low number of TAF can trigger Vδ2 T cell proliferation upon priming with zoledronic acid (ZA) or the antibody drug conjugate Cet-ZA. This effect is similar, although less efficient, to that elicited by monocytes primed with ZA. Herein, we do not focus our attention on the inhibitory effect exerted by stromal cells, but on the possibility that TAF can stimulate immune response when primed with appropriate stimuli. Part of these observations have been added to the Discussion (lines 524-530).

  1. A control group with just T cell and ZA treatment should be added to all the datasets in Figure 1. If this has been done in the previous study, please mention the results and reference in the result description.

We have added the results obtained with T cells and ZA to figure 1 (A-C) as requested. As shown, the priming of T cells with ZA does not trigger the expansion of Vδ2 T cells, in line with our previous reports (ref. 33, 34).

  1. A brief introduction of interaction among monocytes, Vδ2 T lymphocytes and ZA should be added in the Results 3.1. Also, according to the description in Methods, the monocytes seem to be attached the surface during the co-culture. Are they polarized?

In the results section 3.1, a sentence on the interaction among Vδ2 T cells, monocytes and ZA has been added (lines 272-274); also, monocytes have been cited in the introduction (line 68,69). Importantly, monocytes were not present in the co-cultures of T cells and TAF (this is indicated in the supplemental figure 1, upper panels). This is mandatory to define the role of TAF, as also monocytes can trigger Vδ2 T cells proliferation when primed with ZA. In T-monocyte co-cultures, monocytes were not obtained after adherence to plastic, but with a negative selection kit achieving 99% CD3- and 75-84% CD14+ cells (supplemental figure 1, lower panels). It is conceivable that monocytes, upon co-culture with T cells and priming with ZA, have been polarized as M1 macrophages; however, this is not the focus of the present manuscript. Herein, we highlighted the ability of TAF to work similarly to monocytes triggering Vδ2 T cells expansion when primed with ZA or Cet-ZA. This effect occurred in the absence of monocytes in the T-TAF co-cultures.

  1. The representative gating strategy for Fig2A should be shown either in the same figure or in a supplementary figure.

In a supplementary figure (Figure S4) we show the gating strategy used to analyze CD27 and CD45RA expression on Vδ2 T cells, as requested.

  1. The original images from which the cytotoxicity results in Fig 2&4 were generated should be shown in the same figure or in a supplementary figure. Additional cytokine release data should be added to support the cytotoxicity results.

Actually, the crystal violet method to evaluate target cell viability (and indirectly the cytotoxicity) does not imply that images of cultures are taken. Indeed, after the incubation of target and effector cells, the non-adherent cells (dying targets and floating lymphocytes) are harvested and only adherent (living) cells are fixed and stained with crystal violet. After staining, the dye is eluted from cells and spectrophotometric analysis of each well is performed. So, usually no images are shown. Nevertheless, having taken some pictures of cell cultures during our experiments, we have added a supplemental figure that shows the cell cultures stained with crystal violet before the elution (supplemental Figure S2).

We have added data on the production of IFNγ during T:TAF co-cultures in Figure 4H and in the results section (lines 443-446). These data indicate that IFNγ is present in the supernatants from T:TAF co cultures primed with ZA or Cet-ZA. Also, the assay for the evaluation of IFNγ present in culture supernatants have been added in M&M section (lines 225-229).

.

Round 2

Reviewer 2 Report

Accept in present form.